# Thyme and Oregano Oil Potential Therapeutics against Malathion Toxicity through Biochemical, Histological, and Cytochrome P450 1A2 Activities in Male Wistar Rats

**DOI:** 10.3390/ani14192914

**Published:** 2024-10-09

**Authors:** Fatimah A. Al-Saeed, Montaser Elsayed Ali

**Affiliations:** 1Department of Biology, College of Science, King Khalid University, Abha 61413, Saudi Arabia; 2Department of Animal Productions, Faculty of Agriculture, Al-Azhar University, Assiut 71524, Egypt; montaser.elsayd@gmail.com

**Keywords:** cytochrome P450, malathion, thyme essential oil, oregano essential oil, antitoxic

## Abstract

**Simple Summary:**

Pesticides have become essential to control agricultural pests such as weeds, insects, nematodes, bacteria, and fungi, which increases the risk of their transmission to animals through the feed produced from these crops. Among these pesticides, the most widely used is malathion (MOP). It exerts its toxicity through the inhibition of acetylcholinesterase (AChE), an important presynaptic enzyme that terminates nerve impulses by hydrolyzing neurotransmitter acetylcholine. This study highlights the biochemical and histological hazardous reactions of MOP and the detoxifying efficacy of thyme (TEO) and oregano (OEO) essential oils by measuring enzyme-specific activity for Cytochrome P450 1A2 (CYP1A2) in order to reduce environmental hazards. Such knowledge would be the key step toward developing potentially unique treatment options for natural antitoxins. This may allow the livestock sector to increase production and ensure animal welfare and product health and safety.

**Abstract:**

The widespread use of malathion may offer several hazards to humans and animals; additionally, many medicinal plants provide what is known as a broad antitoxicity treatment. This study was carried out to investigate hazardous biochemical and histological reactions to MOP and evaluate the effectiveness of TEO and OEO essential oils in restoring normal physiological conditions after MOP exposure by measuring enzyme-specific activity for Cytochrome P450 1A2 (CYP1A2). One hundred and twenty rats were divided into six groups of twenty animals each: (i) C − MOP served as the control group, (ii) C + MOP treated with 5 mg/kg/BW of Malathion-D10, (iii) TEO treated with 100 mg/kg/BW of oregano essential oil, (iv) TEO treated with 100 mg/kg/BW of thyme essential oil, (v) MOP + OEO treated with 5 mg/kg/BW of Malathion-D10 and 100 mg/kg/BW of oregano essential oil, and (vi) MOP + TEO treated with 5 mg/kg/BW of Malathion-D10 and 100 mg/kg/BW of thyme essential oil. The results indicated the protective effects of OEO and TEO against MOP-induced weight loss. Additionally, there was a significant improvement in ALT, AST, and ALK-Ph after being treated with OEO and TEO, either alone or after MOP exposure. Also, treatment with OEO and TEO ameliorated these oxidative stress parameters, indicating their antioxidative properties. A histopathological examination of liver tissues showed reduced hepatocellular damage and improved liver architecture in the OEO and TEO, both alone and in combination with MOP, and protective effects were more pronounced in the TEO-treated groups. However, the results indicated that TEO was more effective than OEO in increasing CYP1A2 expression and alleviating MOP-induced toxicity. Specifically, TEO showed higher protein expression and therapeutic action in reducing liver damage. In conclusion, these findings suggest that OEO and TEO may be potent therapeutic agents against MOP toxicity, offering protective effects by enhancing CYP1A2 activity and mitigating organ damage. Such knowledge would be an important step toward developing potentially unique treatment options for natural antitoxins.

## 1. Introduction

Since the early 1970s, the organophosphate class of pesticides (OPP) has become the most widely consumed class among pesticides, accounting for almost 50% of the total available insecticides [1], and has been used in the management of plant diseases and crop pests to increase agricultural productivity [2].

Malathion organophosphate (MOP) is one of the OPP groups of pesticides [3], but despite its effectiveness, MOP is associated with potential toxicity risks [4]. The toxicity of MLT is due to its mode of action as an acetylcholinesterase (AChE) inhibitor [5]. The inhibition of AChE causes acetylcholine to accumulate at muscarinic and nicotinic sites, resulting in acute hyperstimulation due to the persistent presence of the neurotransmitter [6].

Animal feedstocks include grass, herbaceous plants, and crops, all of which have a high risk of containing MOP. As each animal can ingest a few kilograms of feedstock pasture per day, upon ingestion, MOP could be retained in the digestive tract, leading to excess exposure to contaminated feed. In addition, workers could be exposed by handling the animals, as well as by cutting, packaging, and transporting MOP-treated by-products [7].

Monitoring studies have been carried out by the FDA to ascertain if MOP residues are present in/on food and feeds; 249 of their samples had MOP concentrations of 0.05–>2.0 ppm [8]. Furthermore, multiple investigations have shown the presence of pesticide residues in animal products after the consumption of contaminated feedstuff [9,10,11].

Cytochrome P450 1A2 (CYP1A2), a key enzyme in the hepatic cytochrome P450 system, plays a crucial role in metabolizing various compounds, including toxins and carcinogens [12]. This enzyme is involved in hydroxylation, DE-alkylation, and oxidative deamination reactions that increase the polarity of substrates [13], facilitating their detoxification [14].

Several essential oils, such as thyme and oregano essential oils (OEO and TEO), originate from herbs native to the Mediterranean region [15]. These essential oils contain phenolic monoterpenes such as thymol and carvacrol, known for their antioxidant properties [16]. The administration of specific natural antioxidants may be an effective method of protecting cellular membranes and biomolecules against oxidative and/or toxic stress-related damage in a biological system [17]. Previous studies have shown that these compounds can induce rat hepatic cytochromes, suggesting potential roles in enhancing detoxification processes [18]. Thyme has been observed to significantly increase the metabolism of substrates for CYP1A, while carvacrol has been found to elevate hepatic CYP3A1 mRNA levels [19]. Given these properties, this study was carried out to affirm the antitoxic claims of OEO and TEO against MOP in male rats.

Although the health of livestock is of great economic importance, there are few studies on the effects of MOP exposure on animals. However, the lack of understanding and awareness regarding its potential effects and the contemporary establishment of MOP application will lead us to expected economic losses. So, this study was carried out to affirm the antitoxic claims of OEO and TEO against MOP. Presenting an experimental therapeutic approach, the current study was aimed at exploring the biochemical, histological, and toxicological effects of these essential oils, focusing on their ability to enhance CYP1A2 activity and restore liver function. Such knowledge may be a step toward developing potentially unique treatment options using natural antitoxins.

## 2. Materials and Methods

### 2.1. Chemicals

Malathion-d10 (diethyl D10, 100 µg/mL^−1^ in acetonitrile, 99.0% isotopic purity; product code: DLM-4476-S; Cambridge, MA 01810, USA) (MLT) was obtained from Kafr El-Zayat Company for Chemicals and Pesticides in Kafr El-Zayat, Egypt, soluble in water and in most organic solvents. On the other hand, TEO and OEO were obtained from Monachem, a strategic partner for specialty chemicals and an Indian-certified contract manufacturer, and their certificates of analysis are provided in Table 1.

### 2.2. Rats and Management

One hundred and twenty male Wistar (*Rattus norvegicus*) rats with a body weight of 134.77 ± 0.69 g and aged 60 d were included in this current study. The animals were purchased from El Osman Farm in Cairo, Egypt. The rats were kept in stainless steel cages and housed under standard conditions of temperature 23 °C and lighting 12 h in light/dark cycles, with free access to food and drinking water ad libitum. The animals were allowed to acclimate to these conditions for two weeks before starting treatment.

### 2.3. Experimental Design

The rats were randomly assigned to six equal groups of 20 rats each: (i) C − MOP served as the control group, (ii) C + MOP treated with 5 mg/kg/BW of Malathion-D10, (iii) TEO treated with 100 mg/kg/BW of oregano essential oil, (iv) TEO treated with 100 mg/kg/BW of thyme essential oil, (v) MOP + OEO treated with 5 mg/kg/BW of Malathion-D10 and 100 mg/kg/BW of oregano essential oil, and (vi) MOP + TEO treated with 5 mg/kg/BW of Malathion-D10 and 100 mg/kg/BW of thyme essential oil. The experimental treatment lasted twenty-one days. Each group was housed in an especially large cage, with a rectangular piece of white paper adhered to the cage for identification. A schematic diagram of the experimental protocol is shown in Figure 1.

### 2.4. Biological Evaluation

The initial weight (IW) and body weight (BW) were evaluated at 7 and 14 days, as well as body weight gain (BWG), calculated using the following formula: BWG = final weight (FW)-initial weight (IW).

### 2.5. Biochemical Assays

A total of 120 blood samples (1 sample × 20 rats × 6 groups) were taken, using a capillary tube to drain the ocular venous plexuses of the rats at the end of the experiment after euthanasia. After centrifuging the blood samples for 20 min at 3000× *g** (gravity force), sera were extracted and stored at −20 °C. Aspartate transaminase (AST) and a significant drop in alanine transaminase (ALT) levels were assayed using BioSystems kits according to Low et al. [20]. Bergmeyer’s guidelines for assaying alkaline phosphatase (ALK-Ph; U/L) were followed by Young [21]. Serum total protein (g/dL) and creatinine (mg/dL) were determined using SPINREACT kits from Chemical Company, Girona, Spain [22]. The colorimetric kinetic approach was used to test glutathione peroxidase (GPx; mU/mL), Butyryl Cholinesterase (U/L), malondialdehyde (MDA; nmol/mL), superoxide dismutase (SOD; U/mL), and total antioxidants (TAC, mM/L), [23].

### 2.6. Histological Examination

At the end of the experimentation, five animals from each group were randomly euthanized to explore potential histological alterations. Specimens from the liver were taken and stored in a neutral buffer with 10% formalin. The samples were dehydrated with increasing alcohol concentrations, cleaned in xylene, embedded in paraffin wax, sliced to 4 μm, stained with hematoxylin and eosin, and viewed under a light microscope [24]. The images are captured with the LABOMED Fluorescence Microscope LX400, cat. no. 9126000, USA. The image scale bar is of 100 µm. The magnification powers are ×100 and ×400.

### 2.7. Specific Gene Detection Technique

The Cytochrome P450 1A2 (CYP1A2) gene was evaluated for pesticide biodegradation capability using oregano and thyme oils. A GeneJET™ Genomic DNA Purification Kit (pub No: MAN0012663; Thermo Fisher Scientific Inc., Waltham, MA, USA) was used to extract and purify the total genomic DNA in accordance with the manufacturer’s instructions.

#### 2.7.1. PCR Amplification

For targeted gene amplification, DreamTaq Green PCR Master Mix (2×; Thermo Fisher, USA) was utilized. Cyp1a2 (NCBI Reference NM_012541.3) was the reference gene used, and specific primers were constructed using the Primer-BLAST program. A total of 10 min of initial denaturation at 94 °C, 35 cycles of denaturation at 94 °C for 1 min, 45 s of annealing at 60 °C, and 1 min of extension at 72 °C were the conditions under which the PCR was carried out. A last extension, which lasted seven minutes at 72 °C, was also provided. The DNA amplification process included an initial denaturation at 94 °C for 10 min, followed by 35 cycles of denaturation at 94 °C for 1 min, annealing at 60 °C for 45 s, and extension at 72 °C for 1 min. A final extension phase was conducted at 72 °C for 7 min to ensure the complete amplification of the target DNA sequences. The primer pairs were selected based on their ability to specifically amplify the CYP1A2 gene with high efficiency. The resulting product lengths were 659 base pairs for primer pair 1 and 445 base pairs for primer pair 4, with melting temperatures ™ and GC content within the optimal range for stable DNA duplex formation (Table 2).

#### 2.7.2. Agarose Gel Electrophoresis

A 1.0% agarose solution was prepared by adding 0.75g agarose to 50 mL of 1x TBE electrophoresis buffer in a 50 mL flask. After heating it in a microwave oven, the agarose was dissolved. The agarose was cooled to 50 °C. A comb was inserted in the electrophoresis bed and the agarose was poured into it. The gel solidified within 15 min and became cloudy; the electrophoresis apparatus (multiSUB Mini, Mini Horizontal Electrophoresis System, Cleaver, Cuyahoga, OH, USA) was filled with the electrophoresis buffer and the comb was removed, creating 6 or 10 wells for sample application in the presence of a DNA ladder (peqGOLD 1 kb DNA-Ladder, Peqlab, VWR), according to manufacturer protocol. Electrodes were connected to the power supply and then later turned on. The energy was adjusted to 80 volts for 100 min. The gel was removed from its bed and transferred to a gel-staining tray for staining with Ethidium bromide for 30 min, followed by a 20 min distain in distilled water. Specific DNA bands were eluted from the agarose gel. Resultant PCR products were purified with an E.Z.N.A.^®^Gel Extraction Kit, (D2500-01, Omega BIO-TEK, Norcross, GA, USA). Sequence analysis was employed using the ABI PRISM^®^ 3100 Genetic Analyzer (Micron-Corp., Seoul, Republic of Korea).

#### 2.7.3. Data Analysis

A gel documentation system (Geldoc-it, UVP, Cambridge, England) was applied for data analysis using Totallab analysis software, ww.totallab.com, (Ver.1.0.1). Aligned sequences were analyzed on the NCBI website (http://www.ncbi.nlm.nih.gov/; accessed on 12 February 2024) using BLAST to confirm their identity. Genetic distances and multialignments were computed via the Pairwise Distance method using Clusteral W (Ver. 1.81) software analysis (http://www.clustal.org/clustal2/; accessed on 2 October 2024). The nucleotide sequences were also compared with sequences available in the GeneBank.

### 2.8. Protein Electrophoretic Studies

SDS-PAGE with 12% resolving gel was performed. After combining liver tissue samples and the loading buffer, 30 µL of the mixture was applied to the gel. For around two hours, electrophoresis was run at 75 volts through the stacking gel and 125 volts through the resolving gel. The gels were immersed in a solution of glacial acetic acid, methanol, and water after being dyed with Coomassie Blue R-250 for two hours.

### 2.9. Statistical Analysis

The submitted data were statistically analyzed using SPSS for Windows 25 (SPSS, Chicago, IL, USA). The Kolmogorov–Smirnov test validated the normal distribution of the data. The data were analyzed using a one-way ANOVA, followed by Duncan’s multiple range tests according to the general linear model Yij = µ + Ti + Aj + Eij, where Yij = experimental observation, µ = general mean, Ti = groups (i = C − MOP, C + MOP, TEO, OEO, MOP + TEO, and MOP + OEO), and eij = experimental error.

## 3. Results

### 3.1. Biological Evaluation

The BW and BWG of the OEO and TEO groups were significantly higher compared to the other groups. However, rats exposed to malathion (C + MOP) alone exhibited a decreased BW and BWG. On the other hand, the groups treated with MOP and essential oils (MOP + OEO and MOP + TEO) showed improvements in BW and BWG compared to the C + MOP, indicating the potential protective effects of OEO and TEO against MOP-induced weight loss (Table 3).

### 3.2. Biochemical Assays

Biochemical assays revealed significant differences in the liver function enzyme (ALT, AST, and ALP), creatinine, and total protein levels between the treated groups and the control group (Table 4). The C + MOP group showed elevated levels of ALT, AST, ALP, and creatinine, indicating liver damage, while the total protein levels were reduced. Treatment with OEO and TEO, both alone and in combination with MOP, improved these parameters, suggesting protective effects against MOP-induced toxicity (Table 4).

### 3.3. Oxidative Stress Markers

Oxidative stress markers, including butyryl cholinesterase, SOD, and MDA, were significantly elevated in rats treated with MOP compared to the control group (Table 5). In contrast, the activity of the GPx enzyme and TAC were significantly decreased. Treatment with OEO and TEO ameliorated these oxidative stress parameters, indicating their antioxidative properties.

### 3.4. Histopathological Examination

The histopathological examination of the rats’ liver tissues showed normal hepatic architecture in the control group (C − MOP; Figure 2A), while the MOP-treated group (C + MOP; Figure 2B) exhibited significant hepatic fibrosis, necrosis, and inflammation. Furthermore, treatment with OEO (Figure 2C) and TEO (Figure 2D), both alone and in combination with MOP (Figure 2E,F), showed reduced hepatocellular damage and improved liver architecture. These protective effects were more pronounced in the TEO-treated groups.

### 3.5. Specific Gene Detection

#### PCR Amplification

The data indicate that the control group (C − MOP) displayed a significant presence of the CYP1A2 gene, as evidenced by the higher lane percentage in band 2 (Table 6 and Figure 3). In contrast, the C + MOP group showed a markedly lower lane percentage, suggesting a decrease in CYP1A2 expression upon malathion exposure. Interestingly, the groups treated with oregano (OEO) and thyme (TEO) essential oils, both alone and in combination with malathion (MOP + OEO and MOP + TEO), exhibited a lane percentage similar to the control, indicating a potential protective effect of these essential oils on CYP1A2 gene expression (Table 6 and Figure 3).

### 3.6. The Identity Matrix Analysis of the CYP1A2 Gene Sequences

The identity matrix analysis of the CYP1A2 gene sequences in male Wistar rats is presented in Table 7. The MOP + TEO group showed slight variations when compared to the other groups, with the lowest identity at 92.99% found in the C + MOP group, which was exposed to MOP only. However, the OEO group, treated with oregano essential oil, displayed the highest sequence identity among all groups, particularly with the reference sequence, highlighting its potential protective effect against genetic changes. The control group (C − MOP) showed high identity with the reference, confirming the stability of the gene sequence under normal conditions. Overall, the essential oils, especially oregano, appear to offer a protective effect, preserving the integrity of the CYP1A2 gene against alterations caused by MOP exposure.

#### 3.6.1. CYP1A2 Sequence

In our study, sequence alignment of the CYP1A2 gene revealed a high degree of conservation across all treatment groups, with the control group (C − MOP) maintaining the reference sequence (Figure 4). Notably, the groups exposed to malathion (C + MOP) and those treated with oregano (OEO) and thyme (TEO) essential oils, both individually and in combination with malathion (MOP + OEO and MOP + TEO), exhibited slight variations. The results underscore the intricate genetic responses elicited by these treatments and highlight the resilience of the CYP1A2 gene sequence to external stressors (Figure 4).

#### 3.6.2. CYP1A2 Phylogenetic Analysis

The phylogenetic analysis of the CYP1A2 gene sequences diverged at various points among groups, indicating different degrees of genetic impact (Figure 5). The branch lengths and bootstrap values suggest that while MOP exposure (C + MOP) led to notable genetic divergence, the essential oils, particularly when combined with malathion (MOP + OEO and MOP + TEO), appeared to mitigate this effect, maintaining closer genetic affinity to the control. This phylogenetic tree underscores the potential of OEO and TEO to preserve genetic stability in the face of MOP stressors, offering a molecular basis for their therapeutic applications.

### 3.7. Protein Electrophoretic Studies

#### 3.7.1. SDS-PAGE Gel Electrophoresis and Molecular Weights (MWs)

The resulting data, summarized in Table 8 and Figure 6, indicate the presence (denoted by 1) or absence (denoted by 0) of protein bands at specific molecular weights (MWs) across the samples. Notably, Band 9 was consistently present across all samples, indicating a common protein component. Similarly, Bands 11, 12, 19, and 22 were universally present, suggesting these proteins are fundamental to the composition of all samples. Certain bands were uniquely expressed in specific sample combinations. For instance, Band 1 was only present in MOP + OEO and MOP + TEO, while Band 4 was absent in all but these two samples, implying a unique protein characteristic in the MOP-combined samples with OEO and TEO. Conversely, Band 2 was exclusive to the C − MOP sample. Other bands showed varied presence across the samples. Band 5 appeared in C − MOP, OEO, TEO, and MOP + TEO but was absent in C + MOP and MOP + OEO. Bands like 10 and 21 were present in specific combinations, providing further differentiation between the samples.

#### 3.7.2. A Visual Tree Phylogenetic Analysis

The tree’s topology indicates that sample MOP + TEO is the most genetically divergent, suggesting a significant evolutionary departure from the common ancestor shared with the other samples (Figure 7). Notable evolutionary events, marked by red indicators on the tree, coincide with key genetic mutations that have contributed to the unique evolutionary path of MOP + TEO. The lengths of the branches imply relative genetic distances, with the longer branches correlating to greater genetic variation. This phylogenetic mapping provides a comprehensive overview of the genetic lineage and affirms the distinctiveness of the evolutionary trajectories within the samples studied.

#### 3.7.3. Concentration Percentages of Cyp1a2 Protein Fractions

The OEO sample exhibited a notably lower concentration at 2.32%, and the TEO sample had a higher concentration of 7.15%. The combination samples, MOP + OEO and MOP + TEO, demonstrated increased concentrations of 11.11% and 14.93%, respectively, suggesting a synergistic effect of MOP with essential oils on Cyp1a2 expression. These results highlight the variable expression levels of Cyp1a2 among the samples, which could be indicative of differing metabolic or detoxification activities within the experimental conditions (Table 9).

## 4. Discussion

This study successfully examined the experimental therapeutic approach of OEO and TEO extracts against MOP intoxication. The essential oils were the most successful candidates among the natural antitoxins [25]. This study revealed significant findings regarding the impact of TEO and OEO on the BW and BWG of rats exposed to MOP. However, this study revealed the toxic effects as disturbances in biochemical as well as histological examination caused by exposure to MOP toxicity. Also, the findings in the present study reported that supplementation with TEO and OEO markedly improved these parameters, suggesting their protective effects against MOP-induced toxicity. Interestingly, the findings indicate a potential protective effect of these essential oils on CYP1A2 gene expression.

The groups treated with both C + MOP and essential oils (MOP + OEO and MOP + TEO) showed improvements in BW and BWG compared to the C − MOP group alone, suggesting a protective effect of OEO and TEO against MOP-induced weight loss [26]. Furthermore, the biochemical assays conducted in this research shed light on the impact of OEO and TEO on liver function enzyme, creatinine, and total protein levels in rats exposed to MOP. MOP exposure led to elevated levels of liver function enzymes (ALT, AST, and ALP) and creatinine, and reduced total protein levels, indicative of liver damage. This result is consistent with earlier research that showed morphologic and clinical changes in parameter measurements after being exposed to MLT [27,28].

Treatment with OEO and TEO, both alone and in combination with MOP, improved these parameters, suggesting protective effects against MOP-induced toxicity [29]. These results highlight the potential of essential oils in preserving liver and kidney function in the face of toxic insults. Moreover, the evaluation of oxidative stress indicators in rats treated with MOP indicated increases in butyryl cholinesterase, SOD, and MDA, as well as reduced activity of the GPx enzyme and TAC. Treatment with OEO and TEO ameliorated these oxidative stress parameters, indicating the antioxidative properties of these essential oils [24,30]. Based on these data, it would seem that EO, with its antioxidant qualities, could enhance antioxidant activity in the treatment groups by preventing the generation of ROS [31,32,33]. These results emphasize the potential of OEO and TEO in combating oxidative stress induced by toxic compounds, thereby safeguarding cellular health and function.

However, after MOP exposure, rat renal tissue shows signs of hypertrophy, structural damage, degenerative changes, inflammation, fibrosis, extravasation, and hemorrhage, indicating chronic inflammation. While macrophages emit pro-inflammatory mediators such as interleukin-1, nitrite oxide, and alpha-tumor [34], as well as a necrosis factor in response to tissue damage, necrotic cells release pro-inflammatory mediators that worsen poison-induced liver injury [35]. Also, the histopathological examination of liver tissues further elucidated the protective effects of OEO and TEO against MOP-induced damage. Treatment with OEO and TEO, both alone and in combination with MOP, resulted in reduced hepatocellular damage and improved liver architecture. These protective effects were more pronounced in the TEO-treated groups, highlighting the efficacy of these essential oils in mitigating hepatic damage [36]. These histopathological findings underscore the therapeutic potential of OEO and TEO in preserving liver integrity and function in toxic environments.

Additionally, the specific gene detection and protein expression analysis conducted in this study provided insights into the molecular mechanisms underlying the protective effects of OEO and TEO. The results indicated that exposure to malathion altered CYP1A2 gene expression, while treatments with OEO and TEO modulated gene expression, suggesting a protective effect of these essential oils [37]. The combined treatments (MOP + OEO and MOP + TEO) further indicated an interaction effect, potentially mitigating malathion’s impact on CYP1A2 expression. These findings highlight the interplay between essential oils and gene expression in response to toxic insults.

## 5. Conclusions

This study demonstrated that exposure to malathion causes toxic effects, including disruptions in biochemical and histological parameters, in addition to the active genetically damaging effect on cytochrome P450 1A2. Also, this study demonstrated that the evaluation of biological, biochemical, oxidative stress, and histopathological parameters, as well as gene detection and protein expression, provides compelling evidence for the protective effects of oregano essential oil (OEO) and thyme essential oil (TEO) against malathion-induced toxicity in rats. The results demonstrate the potential of these essential oils in preserving physiological parameters, mitigating oxidative stress, and modulating gene expression to counteract the detrimental effects of toxic compounds. This research underscores the therapeutic value of OEO and TEO as natural remedies with multifaceted protective properties, offering promising avenues for further exploration in toxicology and pharmacology research.

## Figures and Tables

**Figure 1 animals-14-02914-f001:**
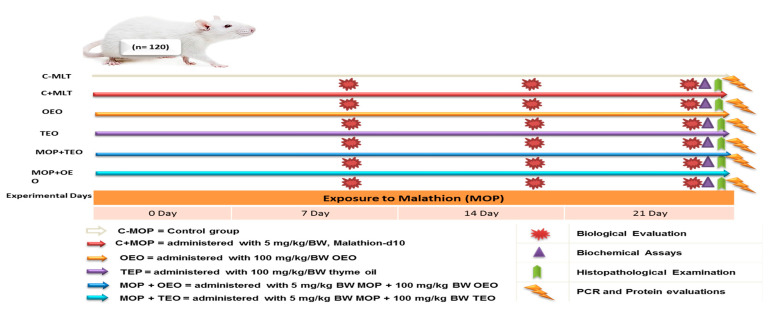
Schematic diagram showing the experimental protocol for thyme and oregano oils’ potentially therapeutic effects against malathion toxicity through biochemical, histological, and cytochrome P450 1A2 activities.

**Figure 2 animals-14-02914-f002:**
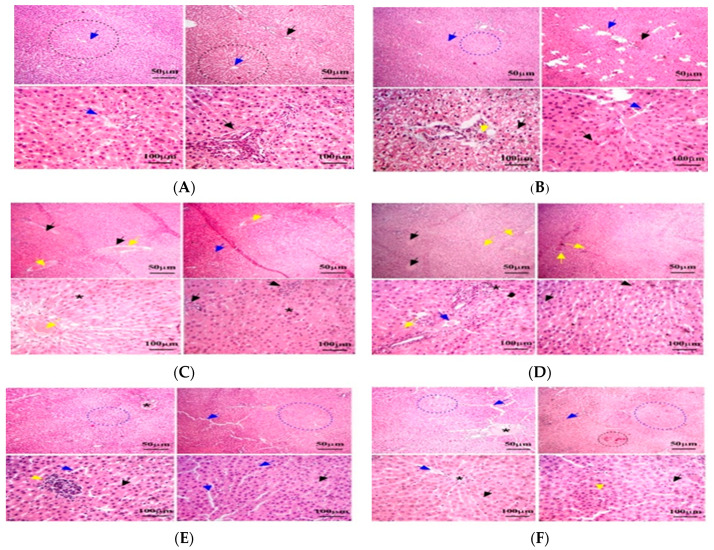
Effect of the thyme and oregano oils as potential therapeutics against malathion toxicity on histopathological changes in liver tissues stained with H&E (magnification ×100 and ×400) in the male rats. (**A**): Liver tissue from C − MOP group showing normal hepatic architecture. (**B**): Liver tissue from C + MOP group showing hepatic fibrosis, necrosis, and inflammation. (**C**): Liver tissue from OEO group showing hepatic steatosis and mild fibrosis. (**D**): Liver tissue from TEO group showing hepatocellular enlargement and mild inflammation. (**E**): Liver tissue from MOP + OEO group showing reduced hepatocellular damage and inflammation. (**F**): Liver tissue from MOP + TEO group showing significant improvement in hepatic architecture and reduced fibrosis. Dotted black circles: normal hepatic nodules. Black arrow: Sinusoids contained numerous Kupffer cells. Yellow arrow: hepatic macrophages. Blue arrow: Bowman’s capsules. Black star: hexagonal lobules, each with a central vein at its core.

**Figure 3 animals-14-02914-f003:**
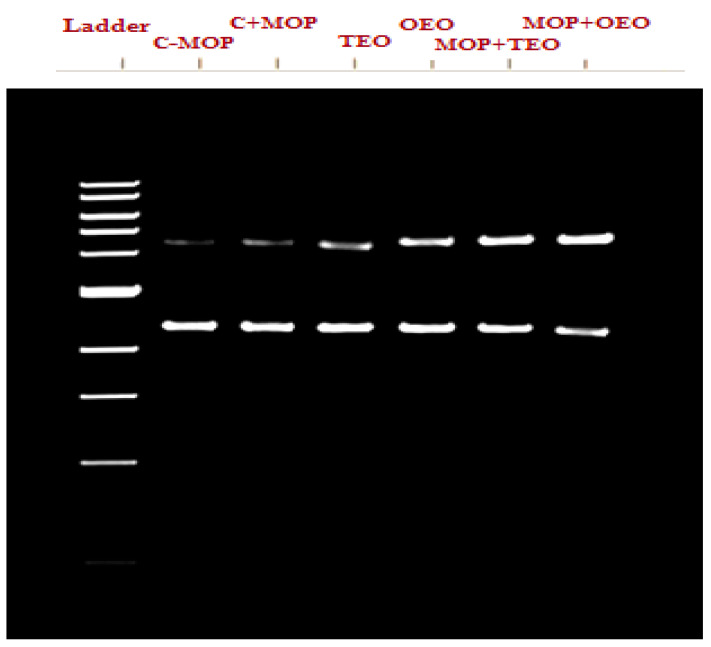
Effect of the thyme and oregano oils as potential therapeutics against malathion toxicity on cytochrome P450 1A2 (CYP1A2) gene expression in the male rats.

**Figure 4 animals-14-02914-f004:**
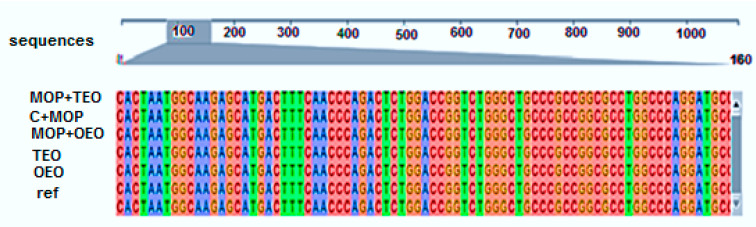
Effect of the thyme and oregano oils as potential therapeutics against malathion toxicity on identity matrix data based on the alignment data of the cytochrome P450 1A2 (CYP1A2) gene sequences in the male rats.

**Figure 5 animals-14-02914-f005:**
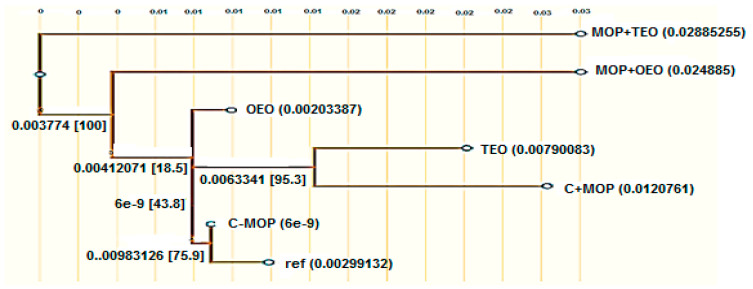
Effect of the thyme and oregano oils as potential therapeutics against malathion toxicity on phylogenetic tree based on cytochrome P450 1A2 (CYP1A2) gene sequences in the male rats. BLAST searches were used to obtain the sequences from the NCBI and UniProt databases. The JTT evolutionary model was used to produce a maximum-likelihood tree, and ancestors were inferred using GRASP’s joint reconstruction technique.

**Figure 6 animals-14-02914-f006:**
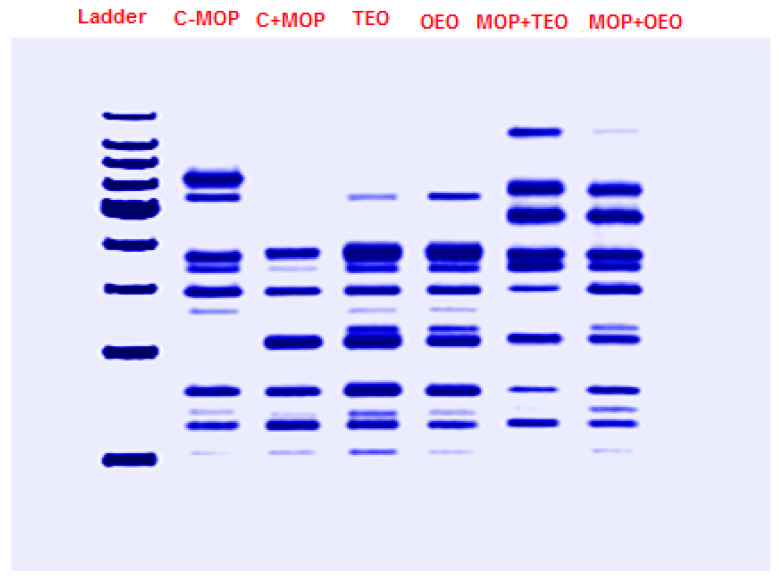
Effect of the thyme and oregano oils as potential therapeutics against malathion toxicity on the phylogenetic tree based on the SDS-PAGE gel electrophoresis results of the protein profiles of the male rats.

**Figure 7 animals-14-02914-f007:**
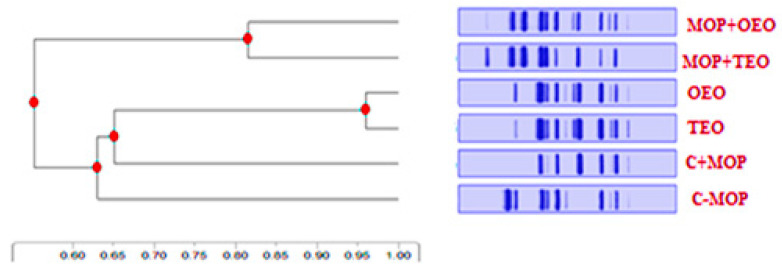
Effect of the thyme and oregano oils as potential therapeutics against malathion toxicity on phylogenetic tree based on visual Cyp1a2 tree phylogenetic analysis of the male rats.

**Table 1 animals-14-02914-t001:** Certificate of TEO and OEO analysis *.

Items	Analysis	
	TEO	OEO
Batch No.	TO/CAL/5021/21-22	
Country of origin	India	
Appearance	Yellow to pale yellow liquid	Yellow- to amber-colored liquid
Odor	The characteristic odor of thyme and sharp, burning taste	Pungent odor, spicy aroma
Solubility in water	Insoluble	Insoluble
Specific gravity	0.919 (0.900–0.930)	0.9370–0.9380
Refraction index at 25 °C	1.4998 (1.4900–1.5100)	1.510–1.520
Content	50.21% (50.00% minimum)	50.85% (50.00% minimum)

* Performed by Monachem, a strategic partner for specialty chemicals and a certified contract manufacturer in India.

**Table 2 animals-14-02914-t002:** Selected primers for Cytochrome P450 1A2 (CYP1A2) gene amplification.

Primers	Sequences (5′ → 3′)	Product Length (bp)	Tm (°C)	GC%
1	GCTGTGGACTTCTTTCCGGT TTTCCCAAGCCGAAGAGCAT	659	60.25 59.96	55.00 50.00
4	CTGAACACCATCAAGCAGGC AGAAGTCCACAGCATTCCCTG	445	59.48 60.00	55.00 52.38

**Table 3 animals-14-02914-t003:** Body weight (g) and body weight gain (g) in the C − MOP, C + MOP, essential oil (OEO and TEO), and therapeutic toxicity (MOP + OEO and MOP + TEO) groups of male Wistar rats.

Items	Initial Weight (0 Day)	BW at 7 Day	BW at 14 Day	Final Weight (21 Day)	BWG
C − MOP	132.67	148.83 ^b^	165.00 ^b^	190.00 ^b^	57.33 ^b^
C + MOP	134.33	142.33 ^c^	151.33 ^c^	161.67 ^c^	27.34 ^c^
OEO	138.33	155.83 ^a^	173.33 ^a^	196.67 ^a^	58.34 ^a^
TEO	133.67	154.83 ^a^	176.00 ^a^	200.00 ^a^	66.33 ^a^
MOP + OEO	136.33	146.00 ^b^	155.67 ^bc^	166.67 ^b^	30.34 ^bc^
MOP + TEO	133.33	147.17 ^b^	160.01 ^b^	181.67 ^b^	48.34 ^b^
SEM	0.64	1.28	2.35	3.82	1.21
*p*-value	0.063	0.001	0.001	0.001	0.001

^a,b,c^ = Duncan test.

**Table 4 animals-14-02914-t004:** Effect of the thyme and oregano oils as potential therapeutics against malathion toxicity on the biochemical parameters of the male rats.

Items	ALT (U/L)	AST (U/L)	ALK-Ph (U/L)	Creatinine (mg/dL)	Total Protein (g/dL)
C − MOP	46.00 ^b^	156.50 ^bc^	107.50 ^cd^	0.57 ^c^	6.90 ^b^
C + MOP	72.00 ^a^	185.50 ^a^	198.00 ^a^	0.93 ^a^	5.20 ^a^
OEO	48.50 ^b^	149.50 ^d^	112.00b ^cd^	0.73 ^bc^	6.30 ^b^
TEO	42.00 ^b^	114.50 ^a^	103.00 ^d^	0.60 ^bc^	6.40 ^b^
MOP + OEO	40.50 ^b^	165.50 ^b^	123.50 ^b^	0.60 ^bc^	6.50 ^b^
MOP + TEO	44.50 ^b^	126.50 ^d^	118.00 ^bc^	0.73 ^b^	6.80 ^b^
SEM	0.59	0.18	0.09	1.02	0.02
*p*-value	0.001	0.001	0.001	0.001	0.001

^a,b,c,d^ = Duncan test.

**Table 5 animals-14-02914-t005:** Effect of the thyme and oregano oils as potential therapeutics against malathion toxicity on the oxidative stress markers of the male rats.

Items	Butyryl Cholinesterase (U/L)	SOD (U/mL)	MDA (nmol/mL)	GPx (mU/mL)	TAC (mM/L)
C − MOP	118.87 ^c^	14.07 ^b^	7.10 ^b^	3.20 ^c^	7.53 ^b^
C + MOP	175.00 ^a^	17.90 ^a^	33.30 ^a^	2.80 ^ab^	2.20 ^c^
OEO	120.00 ^c^	11.17 ^c^	3.67 ^c^	7.30 ^b^	13.33 ^a^
TEO	106.00 ^c^	11.80 ^c^	3.20 ^c^	16.70 ^a^	8.00 ^b^
MOP + OEO	139.50 ^b^	12.90 ^bc^	2.97 ^c^	12.43 ^a^	6.83 ^b^
MOP + TEO	104.00 ^c^	12.47 ^c^	3.87 ^c^	14.90 ^b^	7.83 ^b^
SEM	0.01	0.05	2.78	1.04	0.12
*p*-value	0.01	0.01	0.01	0.01	0.01

^a,b,c^ = Duncan test.

**Table 6 animals-14-02914-t006:** Effect of the thyme and oregano oils as potential therapeutics against malathion toxicity on the computerized detection of cytochrome P450 1A2 (CYP1A2) in the male rats.

Items	Band No.	Lane %	MW (Da)	RF
C − MOP	1	16.16	630.622	0.200
	2	63.93	417.039	0.390
C + MOP	1	0.38	645.509	0.193
	2	0.19	441.637	0.360
OEO	1	0.63	621.370	0.205
	2	63.88	412.966	0.395
TEO	1	0.56	650.781	0.190
	2	63.98	439.812	0.362
MOP + OEO	1	0.53	640.390	0.195
	2	64.10	437.983	0.364
MOP + TEO	1	0.62	667.669	0.183
	2	63.98	437.986	0.364

**Table 7 animals-14-02914-t007:** Effect of the thyme and oregano oils as potential therapeutics against malathion toxicity on identity matrix data based on cytochrome P450 1A2 (CYP1A2) gene sequences in the male rats.

Items	MOP + TEO	C + MOP	MOP + OEO	TEO	OEO	ref	C − MOP
MOP + TEO	100.00	92.99	93.77	93.82	94.90	96.79	96.98
C + MOP	92.99	100.00	94.67	95.60	97.00	95.88	95.88
MOP + OEO	93.77	94.67	100.00	96.00	97.72	95.62	95.62
TEO	93.82	95.60	96.00	100.00	98.12	96.38	96.38
OEO	94.90	97.00	97.72	98.12	100.00	97.75	97.72
ref	96.79	95.88	95.62	96.38	97.75	100.00	99.72
C − MOP	96.98	95.88	95.62	96.38	97.75	99.72	100.00

**Table 8 animals-14-02914-t008:** Effect of the thyme and oregano oils as potential therapeutics against malathion toxicity on the phylogenetic tree based on a data analysis of protein pattern parameters in the male rats.

Band No	MW (kDa)	C − MOP	C + MOP	OEO	TEO	MOP + OEO	MOP + TEO
1	19.299	0	0	0	0	1	1
2	83.144	1	0	0	0	0	0
3	76.173	1	0	0	0	1	0
4	70.107	0	0	0	0	1	1
5	61.582	1	0	1	1	0	1
6	46.572	0	0	0	0	1	1
7	42.039	0	0	0	0	1	1
8	34.683	0	0	1	1	0	0
9	33.436	1	1	1	1	1	1
10	31.850	1	0	0	0	1	1
11	30.242	1	1	1	1	1	1
12	25.564	1	1	1	1	1	1
13	24.523	1	0	0	0	0	0
14	21.852	1	0	1	1	0	0
15	18.362	0	0	1	1	0	1
16	16.694	0	1	0	0	1	1
17	16.095	0	1	1	1	0	0
18	11.907	0	0	1	0	0	0
19	11.589	1	1	1	1	1	1
20	10.747	0	0	1	1	0	1
21	10.580	1	1	0	0	0	0
22	10.363	1	1	1	1	1	1
23	10.067	1	1	1	1	0	1

**Table 9 animals-14-02914-t009:** Effect of the thyme and oregano oils as potential therapeutics against malathion toxicity on the phylogenetic tree based on Cytochrome P450 1A (Cyp1a2) protein fraction concentration (%) in the male rats.

Fractions	C − MOP	C + MOP	OEO	TEO	MOP + OEO	MOP + TEO
Cytochrome P450 1A (Cyp1a2)	9.51	-	2.32	7.15	11.11	14.93

## Data Availability

The raw data supporting the conclusions of this article will be made available by the authors without undue reservation.

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
