# Peer review of "Thyme and Oregano Oil Potential Therapeutics against Malathion Toxicity through Biochemical, Histological, and Cytochrome P450 1A2 Activities in Male Wistar Rats"

_animals, 2024, doi:10.3390/ani14192914_

Round 1

Reviewer 1 Report

Comments and Suggestions for Authors

Dear Authors,

I praise all your efforts trying to show the potential of essential oil in mitigate the toxicity caused by MOP exposure using several experiments. Although, I would recommend a review to be objective and focus only on methods that could really qualify your conclusions. Please see bellow my comments per line.

130- Full name for ALT and AST tests. It’s the first entry on text.

150-Why genomic DNA+PCR to evaluate gene expression instead of complementary DNA (cDNA) using real-time PCR?  DNA has both coding and non-coding sequences; cDNA contains only your coding sequences of interest. Noncoding DNA can disrupt your results on gene expression. Usually, we extract RNA of the sample to get the cDNA and then evaluate by real-time PCR.

210- Correct the text in this line

227-Figures are deformed please resize for a better comparison

230- It was not clear on methods what was the source of DNA, but I assume, tissue or blood. If tissue, I suspect results are more likely linked to figure 2 than a gene expression by itself. In groups where you do not have tissue damage you are more likely to collect more DNA, as consequence, a stronger band on gel. In addition, you are very unlikely to be detecting a real gene expression using the whole DNA and the number of PCR cycles used here.

271- Samples were sequenced according to this section.  No information given on methods and why authors would expect genomic mutations here? It will be surprising to know that whole DNA sample was submitted to sequence the gene here and why would you expect this level of variants in such a short period.  Overall, I would recommend 3.6.1and 3.6.2 removal from the study since there is no evidence that could qualify these results as relevant and useful here.

327-I recommend a review of some conclusions here based on the strength of molecular methods used.

353-English correction

371-It was not specific gene detection

Comments on the Quality of English Language

Overall it was well written and understandable.

Author Response

We appreciate you for your precious time in reviewing our manuscript. We have carefully considered your comments and tried our best to address them to make an extensive revision. We hope the manuscript's careful revisions meet your high standards. Our detailed, point-by-point responses to the comments are given below.

  • Journal: Animals(ISSN 2076-2615).
  • Manuscript ID: animals-3200187
  • Type of manuscript: Article
  • Title: Thyme and oregano oils potential therapeutic against malathion toxicity through biochemical, histological, and Cytochrome P450 1A2 activities
  • Authors: Fatimah A. Al-Saeed *, M E Ali
  • Received: 30 Aug 2024

# - Comments and Suggestions for Authors

Comments 1: [I praise all your efforts trying to show the potential of essential oil in mitigate the toxicity caused by MOP exposure using several experiments. Although, I would recommend a review to be objective and focus only on methods that could really qualify your conclusions. Please see bellow my comments per line].

Response 1: [Thank you very much for your supportive comment. We appreciate the time and effort you took to review our manuscript and we are doing our best to improve the quality of the manuscript with your valuable requirements].

Comments 2: [130- Full name for ALT and AST tests. It’s the first entry on text].

Response 2: [Thank you very much for your comment. ALT and AST full name were added].

Comments 3: [150-Why genomic DNA+PCR to evaluate gene expression instead of complementary DNA (cDNA) using real-time PCR? DNA has both coding and non-coding sequences; cDNA contains only your coding sequences of interest. Noncoding DNA can disrupt your results on gene expression. Usually, we extract RNA of the sample to get the cDNA and then evaluate by real-time PCR.].

Response 3: [Thank you very much for your valuable comment which comes from high experience and which we can work on later. We built this application based on previous scientific theories: A PCR template for replication can be of any DNA source, such as genomic DNA (gDNA), complementary DNA (cDNA), and plasmid DNA. Nevertheless, the composition or complexity of the DNA contributes to optimal input amounts for PCR amplification [1]. We will work hard on the modern advanced study of research to reach high quality to meet your valuable requirements in subsequent works.].

Comments 4: [210- Correct the text in this line].

Response 4: [Thank you very much for your comment, the text has been corrected for both the title of the tables and figures throughout the manuscript.].

Comments 5: [227-Figures are deformed please resize for a better comparison].

Response 5: [Thank you very much for your comment, we have retrieved all the forms from the original file, hope they are acceptable for your valuable requirements].

Comments 6: [230- It was not clear on methods what was the source of DNA, but I assume, tissue or blood. If tissue, I suspect results are more likely linked to figure 2 than a gene expression by itself.In groups where you do not have tissue damage you are more likely to collect more DNA, as consequence, a stronger band on gel. In addition, you are very unlikely to be detecting a real gene expression using the whole DNA and the number of PCR cycles used here.].

Response 6: [Thank you very much for your valuable comments that added a lot of knowledge to us. The source was from liver tissue samples collected at the end of the experiment from the mice. In addition, we agree with your valuable comment as we indicated in the results that the control group which was histologically healthy displayed a significant presence of the CYP1A2 gene, as evidenced by the higher lane percentage in band 2. In contrast, the C+MOP group showed a significantly lower lane percentage, suggesting a decrease in CYP1A2 expression upon Malathion exposure].

Comments 7: [271 Samples were sequenced according to this section.  No information given on methods and why authors would expect genomic mutations here? It will be surprising to know that whole DNA sample was submitted to sequence the gene here and why would you expect this level of variants in such a short period.  Overall, I would recommend 3.6.1and 3.6.2 removal from the study since there is no evidence that could qualify these results as relevant and useful here].

Response 7: [We thank you very much for your comment; we have provided the information required for data analysis in section 2.3.7. Based on the data analysis, no genetic mutations were observed and we performed this analysis to give a clearer picture of the treatment used and to compare groups. We are ready to implement any comments that improve the quality of the manuscript, whether by adding or removing parts as deemed appropriate by the editors and ].

Comments 8: [I recommend a review of some conclusions here based on the strength of molecular methods used.].

Response 8: [Thank you very much for your valuable comment, we have edited the conclusion based on the points mentioned and offer you more appreciation].

Comments 9: [353-English correction].

Response 9: [Thank you very much for your comment, the sentence has been edited and corrected].

Comments 10: [It was not specific gene detection].

Response 10: [Thank you very much for your comment. The detection of the cytochrome gene was one of the targets of this study and we worked to link its activity to the stress resulting from malathion toxicity and its activity under treatment with essential oils as a natural antoxic.].

Comments 11 [Comments on the Quality of English Language], [Overall it was well written and understandable].

We appreciate your kind time in reviewing and editing the manuscript and improving its quality and we hope that we have met your valuable requirements.

[1] https://www.thermofisher.com/eg/en/home/life-science/cloning/cloning-learning-center/invitrogen-school-of-molecular-biology/pcr-education/pcr-reagents-enzymes/pcr-component-considerations.html

Reviewer 2 Report

Comments and Suggestions for Authors

General Comments:

·       Lack of Dose-Response Analysis: The study employed fixed doses of MOP, OEO, and ThEO without investigating how varying doses could influence outcomes. A dose-response study could provide more insights into the safety and efficacy of these treatments at different levels of exposure.

·       The author used a low MOP dose; for this, a longer study would provide more data on the chronic impacts of MOP and the long-term protective effects of OEO and ThEO.

·       Why were there no Kidney or brain histopathological results? MOP toxicity also involves these vital organs.

Title: The title should show that this research was performed in rats.

The study used rats as models are commonly used, there are often significant differences in biochemical pathways between species.

Abstract

Line23-25:

Efficiency is not correct as it assumes the research hypothesis, I suggest: evaluating the effectiveness of thyme and oregano essential oils in restoring normal physiological conditions.

Line 33-34: Additionally, treatment with OEO and TEO, both alone and in combination with....MOP, improved liver function. Not clear.

Line 39: Are there any clinical signs of MOP toxicity noted? If yes please mention. If not what is your explanation?

Introduction

Line 47: OPC or OPP?

Line 52: MLT or MOP.

Materials and Method

Line 93: MLT or MOP.

Tabl1 is not needed in this study. Mention the manufacturer or source.

Line 102: more details about breed and type of animals used.

Line 124: change to: The initial weight (IW) and body weight (BW) were evaluated…..

Line 125: what is FW?

Line 129: rpm is no longer used, please convert to g* (Gravity force).

Line 130: ALT and AST add abbreviations.

Line 151-165: Why there were 2 PCR conditions? Is it Real Time Taq profile? Not clear.

Line 167-175: Is it possible to run 10% agarose for 20 min to get Fig 3? Please check agarose concentration and add the details in Fig 3 caption.  

Line 177: What were the samples for SDS?

Results

Line 197: data from Table 3 can presented in figures with bars and SEM rather than a table.

Line 208 and Line 217: Same as Table 3.

Line 228-243: How could the author argue the results of PCR from the gel? This should be analyzed by Real-Time PCR Ct normalized to references or housekeeping genes.

Line 268: What is the significance of Fig 4?

Line 271: CYP1A2 phylogenetic analysis. Not mentioned in materials and method.

Line 281: Mention the methodology of the tree construction the software used and the bootstrapping method selected.

Discussion:

Line 351: SOD, MDA and other abbreviations were missing.

The study focuses on the antioxidative and hepatoprotective effects of OEO and ThEO. Still, it does not explore other potential mechanisms, such as anti-inflammatory or immunomodulatory properties, that may also play a role in counteracting MOP toxicity.

Comments on the Quality of English Language

Minor editing of English language required.

Author Response

We appreciate you for your precious time in reviewing our manuscript. We have carefully considered your comments and tried our best to address them to make an extensive revision. We hope the manuscript's careful revisions meet your high standards. Our detailed, point-by-point responses to the comments are given below.

  • Journal: Animals(ISSN 2076-2615).
  • Manuscript ID: animals-3200187
  • Type of manuscript: Article
  • Title: Thyme and oregano oils potential therapeutic against malathion toxicity through biochemical, histological, and Cytochrome P450 1A2 activities
  • Authors: Fatimah A. Al-Saeed *, M E Ali
  • Received: 30 Aug 2024

# - Comments and Suggestions for Authors

Comments 1: [Lack of Dose-Response Analysis: The study employed fixed doses of MOP, OEO, and ThEO without investigating how varying doses could influence outcomes. A dose-response study could provide more insights into the safety and efficacy of these treatments at different levels of exposure].

Response 1: [Thank you very much for your valuable comment, we have determined the doses based on previous studies, and we appreciate your valuable requirements. We will work hard to delve into a dose-response study which could provide more insights into the safety and efficacy of these treatments at different levels of exposure in a future project where we are study the effects of the agents that act as antitoxic].

Comments 2: [The author used a low MOP dose; for this, a longer study would provide more data on the chronic impacts of MOP and the long-term protective effects of OEO and ThEO].

Response 2: [Thank you very much for your valuable comment, we were very cautious in using the half-lethal dose of malathion according to previous studies "Malathion-d10 (diethyl D10, 100 µg/mL−1 in acetonitrile, 99.0% isotopic purity; product code: DLM-4476-S; Cambridge, MA 01810, USA)." for a period of 3 weeks and according to previous studies is the period after which death occurs and in our study we worked hard to show the therapeutic role of the oils and we concluded that we need more study to determine the safe doses, please note that we tried to study the anti-toxic role of the oils, we greatly appreciate your valuable and constructive comment, thank you very much.].

Comments 3: [Why were there no Kidney or brain histopathological results? MOP toxicity also involves these vital organs].

Response 3: [Thank you very much for your valuable and thoughtful comment. This study included only liver histology in order to link it to the genetic study of Cytochrome P450 1A2 activities studied from liver tissue as well. We look forward to future studies that will clarify more aspects.].

Comments 4: [Title: The title should show that this research was performed in rats.].

Response 4: [Many thanks for your valuable editing of the manuscript title. The title has now been edited to "Thyme and Oregano Oils Potential Therapeutic against Mala-thion Toxicity through Biochemical, Histological, and Cytochrome P450 1A2 Activities in Wistar male rats."].

Comments 5: [The study used rats as models are commonly used; there are often significant differences in biochemical pathways between species.

Response 5: [Thank you very much for your valuable and effective education; we appreciate your study and your valuable comment. Mice are often used as a model for experiments to apply many treatments to be gradually applied to other animals. On the other hand, rats have contributed the lifeblood of all living species, including humans, animals, and plants, by offering a wide range of therapeutic services and vaccines that provide radical remedies to a variety of issues and diseases. While Mice and rats make up approximately 95% of all laboratory animals, with mice the most commonly used animal in biomedical research. Mice are a commonly selected animal model. Mice have been used as research subjects for studies ranging from biology to psychology to engineering. They are used to model human diseases for the purpose of finding treatments or cures. Some of the diseases they model include: hypertension, diabetes, cataracts, obesity, seizures, respiratory problems, deafness, Parkinson's disease, Alzheimer's disease, various cancers, cystic fibrosis, and acquired immunodeficiency syndrome (AIDS), heart disease, muscular dystrophy, and spinal cord injuries. Mice are also used in behavioral, sensory, aging, nutrition, and genetic studies. This list is in no way complete as geneticists, biologists, and other scientists are rapidly finding new uses for the domestic mouse in research. Furthermore, all Institutional and National Guidelines for the care and use of animals were followed in accordance with the Egyptian Medical Research Ethics Committee (no. 14-126), and the Research Ethics Committee of the Faculty of Agriculture at Assiut University granted ethical approval for the aforementioned research project (Reference No: 03-2024-0007)].

Comments 6: [Line23-25: Efficiency is not correct as it assumes the research hypothesis, I suggest: evaluating the effectiveness of thyme and oregano essential oils in restoring normal physiological conditions.].

Response 6: [Thank you very much for your valuable education, the sentence has been edited and corrected.].

Comments 7: [Line 33-34: Additionally, treatment with OEO and TEO, both alone and in combination with....MOP, improved liver function. Not clear].

Response 7: [Thank you very much for your valuable education, the sentence has been edited and corrected].

Comments 8: [Line 39: Are there any clinical signs of MOP toxicity noted? If yes please mention. If not what is your explanation?].

Response 8: [Thank you very much for your valuable comment. In this part of the study, the aim was to evaluate the efficiency of OEO and TEO oils after MOP. We relied on tissue sections from the liver for histological and genetic examination, and did not address any other histological or clinical examination.].

Comments 9: [Introduction "Line 47: OPC or OPP?"].

Response 9: [Thank you very much for your valuable comment, the first two letters refer to organophosphate and the third letter refers to pesticide and the S is added to indicate the plural while we used the abbreviation MOP for organophosphate.].

Comments 10: [Introduction " Line 52: MLT or MOP"].

Response 10: [Thank you very much for your comment, the sentence has been edited and corrected. We have used MOP as an abbreviation for the type of malathion used "Malathion organophosphate (MOP)"].

Comments 11: [Line 93: MLT or MOP].

Response 11: [Thank you very much for your comment, the sentence has been edited and corrected. We have used MOP as an abbreviation for the type of malathion used "Malathion organophosphate (MOP)"].

Comments 12: [Tabl1 is not needed in this study. Mention the manufacturer or source.].

Response 12: [Thank you very much for your valuable and constructive comment. We are fully prepared to edit any part of the manuscript as the reviewers and editors see fit. We have left it for now to avoid any conflict between the reviewers' views and we are fully prepared to implement what they see fit to improve the quality of the manuscript.].

Comments 13: [Line 102: more details about breed and type of animals used.].

Response 13: [Thank you very much for your valuable constructive comment, data about breed and type of animals].

Comments 14: [Line 124: change to: The initial weight (IW) and body weight (BW) were evaluated…..].

Response 14: [Thank you very much for your valuable education, the sentence has been edited and corrected].

Comments 15: [Line 125: what is FW?].

Response 15: [Thank you very much for your valuable comment, please accept my apologies for this loss. The equation has been edited: "BWG = Final Weight (FW) - Initial Weight (IW)."].

Comments 16: [Line 129: rpm is no longer used, please convert to g* (Gravity force).].

Response 16: [Thank you very much for this knowledge it has been edited and fixed.].

Comments 17: [Line 130: ALT and AST add abbreviations.].

Response 17: [Thank you very much for your valuable comment. The full name of the abbreviation has been provided.].

Comments 18: [Line 151-165: Why there were 2 PCR conditions? Is it Real Time Taq profile? Not clear.].

Response 18: [Thank you very much for your valuable comments which have enriched the quality of the manuscript. According to the manufacturer's instructions, tow cytochrome primers were designed as shown in Table 2, as a selected primers for Cytochrome P450 1A2 (CYP1A2) gene amplification.

Table 2. Selected primers for Cytochrome P450 1A2 (CYP1A2) gene amplification.

Primers

Sequences (5’ → 3’)

Product length (bp)

Tm (°C)

GC%

1

GCTGTGGACTTCTTTCCGGT

TTTCCCAAGCCGAAGAGCAT

659

60.25

59.96

55.00

50.00

4

CTGAACACCATCAAGCAGGC
AGAAGTCCACAGCATTCCCTG

445

59.48

60.00

55.00

52.38

For targeted gene amplification, DreamTaq Green PCR Master Mix (2X; Thermo Fisher, USA) was utilized. Cyp1a2 (NCBI Reference NM_012541.3) was the reference gene used, and specific primers were constructed using the Primer-BLAST program. Ten minutes of initial denaturation at 94 °C, 35 cycles of denaturation at 94 °C for one minute, 45 seconds of annealing at 60 °C, and one minute of extension at 72 °C were the conditions under which the PCR was carried out. A last extension, which lasted seven minutes at 72 °C, was also provided.].

Comments 19: [Line 167-175: Is it possible to run 10% agarose for 20 min to get Fig 3? Please check agarose concentration and add the details in Fig 3 caption.  ].

Response 19: [Thank you very much for your careful review and valuable comments. We have reviewed the source and found that there was an unintended mistake in transferring the sentence to the manuscript. We have edited and corrected it and described the process accurately. Thank you for this favors and please accept our apologies in this regard.].

Comments 20: [Line 177: What were the samples for SDS?].

Response 20: [Thank you very much for your valuable comment, the text has been edited to "SDS-PAGE with 12% resolving gel was performed. After combining liver samples tissue and loading buffer, 30 µl of the mixture was applied to the gel. "].

Comments 21: [Line 197: data from Table 3 can presented in figures with bars and SEM rather than a table.].

Response 21: [Thank you very much for your valuable and constructive comment. We are fully prepared to edit any part of the manuscript as the reviewers and editors see fit. We have left it for now to avoid any conflict between the reviewers' views and we are fully prepared to implement what they see fit to improve the quality of the manuscript.].

Comments 22: [Line 208 and Line 217: Same as Table 3.].

Response 22: [Thank you very much for your valuable and constructive comment. We are fully prepared to edit any part of the manuscript as the reviewers and editors see fit. We have left it for now to avoid any conflict between the reviewers' views and we are fully prepared to implement what they see fit to improve the quality of the manuscript.].

Comments 23: [Line 228-243: How could the author argue the results of PCR from the gel? This should be analyzed by Real-Time PCR Ct normalized to references or housekeeping genes.].

Response 23: [Thank you very much for your valuable comment, we have followed your valuable instructions and hope we have met your requirements.].

Comments 24: [Line 268: What is the significance of Fig 4?].

Response 24: [Thank you very much, the Fig. 4 is CYP1A2 sequence, sequence alignment of the CYP1A2 gene revealed a high degree of conservation across all treatment groups, with the control group (C-MOP) maintaining the reference sequence. We are fully prepared to edit any part of the manuscript as the reviewers and editors see fit.].

Comments 25: [Line 271: CYP1A2 phylogenetic analysis. Not mentioned in materials and method.].

Response 25: [Thank you very much for your constructive teaching, it has been provided YP1A2 phylogenetic analysis in materials and method section.].

Comments 26: [Line 281: Mention the methodology of the tree construction the software used and the bootstrapping method selected.].

Response 26: [Thank You very much for your valuable comment, your constructive requirements has been met.].

Comments 27: [Line 351: SOD, MDA and other abbreviations were missing.].

Response 27: [Thank you very much for your valuable comment, we have defined the abbreviations in the Study Materials and Methods section.].

Comments 28: [The study focuses on the antioxidative and hepatoprotective effects of OEO and ThEO. Still, it does not explore other potential mechanisms, such as anti-inflammatory or immunomodulatory properties, that may also play a role in counteracting MOP toxicity.].

Response 28: [Thank you very much for your valuable and constructive comment, which may open many other future studies for us. This study shed light on the dangerous biochemical and tissue interactions of MOP and the detoxification efficacy of essential oils of thyme (TEO) and wild thyme (OEO) by measuring the specific enzyme activity of Cytochrome P450 1A2 (CYP1A2) in order to reduce environmental risks. This knowledge will be the main step towards developing potential unique treatment options for natural antidotes. This opened many other studies for clinical studies and the use of Dose-Response Analysis levels to reach new horizons].

Comments 29: [Comments on the Quality of English Language

Minor editing of English language required.].

Response 29: [We appreciate your kind time in reviewing and editing the manuscript and improving its quality and we hope that we have met your valuable requirements.].

Round 2

Reviewer 2 Report

Comments and Suggestions for Authors

The authors responded properly and sufficiently to the commentaries.